# Quarantine supervision of Wood Packaging Materials (WPM) at Chinese ports of entry from 2003 to 2016

Jiaqiang Zhao[1], Ke Hu[2], Ke Chen[3], Juan Shi[1]*

1 Sino-French Joint Laboratory for Invasive Forest Pests in Eurasia, College of Forestry, Beijing Forestry University, Beijing, China, 2 Criminal Investigation Corps, Beijing Municipal Public Security Bureau, Beijing, China, 3 Animal and Plant Quarantine Institute, Chinese Academy of Inspection and Quarantine, Beijing, China

* shi_juan@263.net

## Abstract

Exotic pests have caused huge losses to agriculture, forestry, and human health. Analyzing information on all concerned pest species and their origin will help to improve the inspection procedures and will help to clarify the relative risks of imported cargo and formulate international trade policies. Records of intercepted pests from wood packaging materials (WPM) from 2003 to 2016 in the China Port Information Network (CPIN) database were analyzed. Results showed that the number of intercepted pests from WPM was lowest in the first quarter and highest in the fourth one. The total number of interceptions increased each year, with 53.33% of intercepted insects followed by nematodes (31.54%). The original continent of most intercepted pests was Asia (49.29%). Xylophagous insects were primarily intercepted from Southeast Asian countries, whereas nematodes were primarily intercepted from Korea, Australia, Mexico, and other countries. WPM interception records were mainly concentrated in China's coastal inspection stations (98.7%), with the largest number of interceptions documented in Shanghai, followed by the inspection stations of Jiangsu Province. The proportion of pest taxa intercepted by the Chinese provinces' stations each year is becoming increasingly balanced. The number of pest disposal treatment measures for intercepted cargoes with dead non-quarantine pests increased significantly from 2012 to 2016. This reflects the fact that Chinese customs inspection stations are becoming increasingly scientific and standardizing the interception and treatment of WPM pests. The issues reflected in the database, with a view to providing a reference for future work by customs officers and researchers.

## Introduction

Due to technological progress and global trade, goods and products are flowing around the world at an ever-increasing speed and frequency. This movement has led to a substantial increase in biological invasions by allowing organisms to pass through natural barriers that

**Data Availability Statement:** All relevant data are within the manuscript and its files.

**Funding:** This study was supported by the General Program of National Natural Science Foundation of China (31770687) and Forestry Science and

Technology Innovation Special of Jiangxi Forestry Department (201912).

**Competing interests:** The authors declare no conflicts of interest.

typically limit their spread [1–6]. China has a vast territory with a diverse climate. Its planted forest area ranks first in the world, and the volume of international trade has increased from year to year. Therefore, China is among the countries most severely affected by foreign pests [7–9].

According to a survey conducted by the Food and Agriculture Organization of the United Nations (FAO), about 70% of the goods imported and exported between countries use wood packaging materials (WPM) [10]. Because WPM do not reflect the value of goods in the trade process, inferior wood is primarily used as a raw material [11]. WPM that have not undergone effective pest control treatment often carry multiple pests that can appear on the surface (e.g. bark beetles, moths, fungi, etc.) or inside the wood (e.g. boring insects, nematodes, fungi, etc.) [12, 13] or were traded [14, 15]. Inspection data from the United States and New Zealand showed that WPM, including crates, pallets, and dunnage, are the most common high-risk sources of bark beetles, woodborers, and wilt or stain fungi [16, 17]. If these pests successfully colonize and multiply after arriving at the destination port, they pose a serious threat to the agricultural and forest ecological security of the destination country. Therefore, inspection and pest control of entering WPM have become a focus of quarantine departments in various countries [18, 19].

At present, most researchers analyze WPM pest interception data only for bark- and wood-infesting insects. In the United States, Haack conducted a systematic analysis of Coleoptera in WPM pest [20] and Mccullough *et al.* [21] analyzed interception data for nonindigenous plant pests for 17 years and found that within specific commodity pathways, richness of the pest taxa generally increased linearly with the number of interceptions. In China, the main quarantine pests in imported WPM are insects and nematodes [22]. Platypodidae, Scolytidae, Cerambycidae, and two other families of insects and nematodes (mainly the pinewood nematode) [22]. Xia *et al.* [23] analyzed the annual trends, population types, and interception frequencies of quarantine pests intercepted on imported wood packaging in Shandong Province, China.

The imported WPM epidemic situation is closely related to the country (region) of origin and intercepted batches. Types of pests vary significantly among countries (regions). Therefore, quarantining entering WPM should be performed at the source of imported WPM epidemics, so as to propose more targeted quarantine strategies [24, 25]. Analyzing intercepted borers or quarantine pests on WPM will greatly underestimate the living organism groups and quantities of pests they carry, thereby minimizing the real harm caused by imported WPM [23].

In this context, the main purpose of the present study was to use WPM interception data from 2003 to 2016 to: (i) systematically describe the main source countries (regions) of inbound WPM in China; (ii) analyze the overall characteristics of intercepted pests from different countries (regions) and in different provinces, and (iii) make a preliminary assessment of China's current port WPM quarantine. This analysis provides a background dataset and scientific rationale for managing future WPM quarantine of incoming goods at Chinese ports and will help to prevent foreign pests from entering China on WPM through international trade.

## Materials and methods

### Sampling

The intercepted WPM pest data used in this article were downloaded from the China Port Information Network (CPIN) database. Due to the complexity of the database, possible misunderstandings, and even international trade disputes, these data are rarely published. Each entry

records details such as the CIQ code, country of origin, date of reporting, immediate bureau, shipping carrier, scientific name, survival status, handling measures, etc.

The CPIN has recognized limitations. The sampling of goods is based on a risk assessment of the type of goods, country (region) of origin, company qualifications, and other information used by Chinese customs to formulate corresponding cargo sampling instructions, rather than on random sampling. The data record information only on shipments in which pests were found, and there is no record of goods without interception. The data are not statistically robust, so only a small number of statistical tests can be performed. Due to time constraints or the inaccessibility (contact, proximity) of some shipments, the number or frequency of intercepted pests in one shipment is usually not recorded, and the discovery of an actionable pest usually leads to regulatory action, thereby avoiding the need for further inspection [26].

A total of 464,512 wood packaging interception records from January 1, 2003 to December 31, 2016 were used for analysis and were divided into five groups: Insects (I), Nematodes (N), Weeds (W), Pathogens (P), and Others (O) (mites, spiders, mollusks, etc.). Records were carefully checked to correct typing or typesetting errors, and lists of synonyms were compiled for all species to prevent duplicate records [27].

## Statistical analysis

CPIN data were queried and cross-indexed using Microsoft Access to obtain initial statistics on overall interceptions of wood packaging pests, source states, source countries (regions), and interceptions at Chinese ports. To further study the interception of pests on wood packaging from various countries (regions), cluster analysis was carried out using SPSS 22.0 software, and Ward's systematic clustering method. Four statistical variables were used: (X1) the total species of intercepted pests; (X2) the total number of intercepted pests; (X3) the interception rate of quarantine pests (quarantine pests intercepted/X2); and (X4) the interception rate of insects and nematodes (insects and nematodes intercepted/X2).

Factorial and correspondence analyses were carried out using R 4.0.3 software to analyze the country (region) distribution of nematodes, xylophagous insects (Cerambycidae, Scolytidae, Platypodidae, and Bostrichidae), and storage pests, which are of high concern and frequently intercepted on WPM.

Cytoscape 3.7.1 software was used to construct a "survival status-quarantine status-treatment measures" visualization network for wood packaging interception data from China's ports and to conduct network topology analysis.

## Results and discussion

### Wood packaging of goods imported to China

The interception data encompasses goods from six continents, and the number of pest species intercepted in the wooden packaging of imported goods shows an increasing yearly trend (Fig 1). This increasing trend can be explained by increased trade volume and better awareness, effectiveness, skills, and detection methods of customs officers [28–30]. The number of pests intercepted each year is positively correlated with the total import trade ($R = 0.91$, $N = 14$, $P<0.01$). Total import trade declined in 2009 and 2015, mainly due to the global economic crisis and the economic downturn in those years, which had a negative impact on interceptions [31, 32].

Among the intercepted pests, 42.59% were identified to the family level, 26.86% to the genus level, and 22.16% to the species level. In Australia, a similar degree of identification was reported [33]. Live pests accounted for 273,138 interceptions (58.8% of all records), and quarantine pests accounted for 19,590 interceptions (4.22%). From 2003 to 2016, there were 33,179

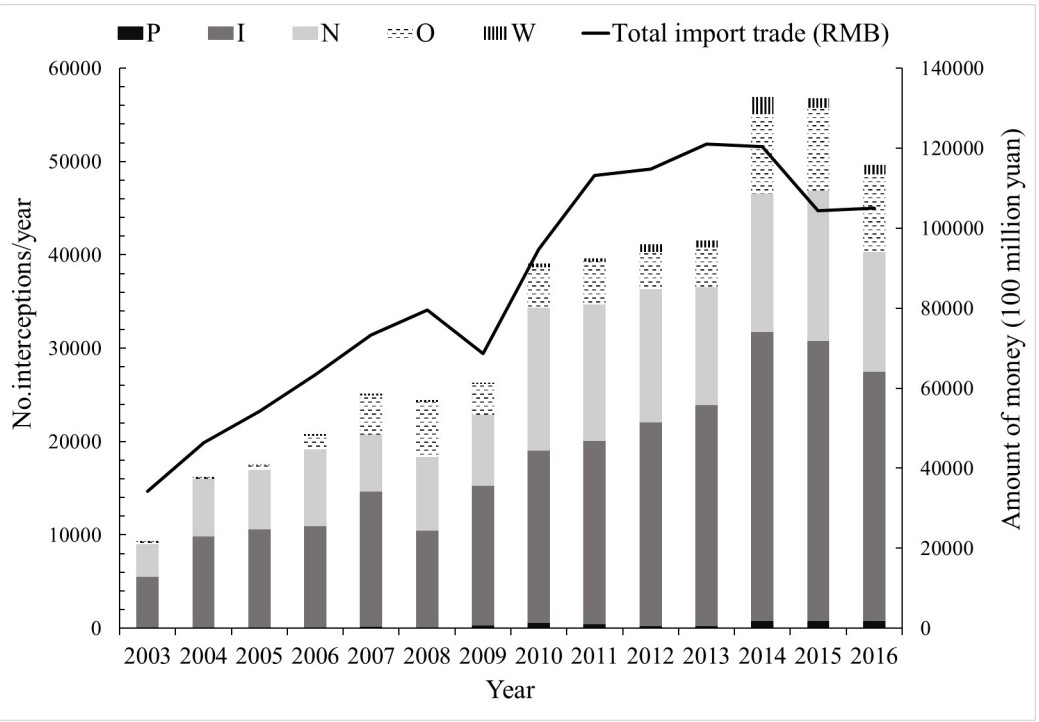

**Fig 1. Number of intercepted pests and total import trade from 2003 to 2016.** I = Insects, N = Nematodes, W = Weeds, P = Pathogens, O = Others (e.g. mites, spiders, mollusks).

insects (SE±71.96%) intercepted per year on average, and the number of species intercepted increased by 15.83% (SE±58.69) per year on average. Insects and nematodes accounted for 53.33% and 31.54% of all records, respectively, followed by other species (12.59%), weeds (1.57%), and pathogens (0.96%). Weed interceptions had the greatest multiplicative increase over the 14 years. In 2014, the number of weeds (1,842 species) was 76.75 times higher than in 2003. Insect interceptions had the largest increase in absolute numbers, with 25,479 more insect interception records in 2014 than in 2003 (Fig 1).

## Continent of origin

A total of 20,378 (4.39%) of all records were removed because they indicated "country of unknown origin". There are differences in the taxa intercepted from different source continents: insects predominate in goods from South America and Africa, whereas insects and nematodes together predominate in goods from other continents (Fig 2). The largest numbers of intercepted pests derived from Asia (49.29%), Europe (26.90%) and North America (13.74%). The lowest ones were recorded in South America (3.54%), Oceania (1.37%), and Africa (0.77%) (Fig 2). In Asia, most interceptions occurred in goods from East Asian (66.66% of Asian records) and Southeast Asian (25.9%) countries, and interceptions from Europe were dominated by those from Central (48.22% of European records), Western (25.38%), and Southern European countries (17.93%). Quarterly interceptions from each continent of origin show a clear cycle, with the fewest interceptions in the first quarter of each year, more interceptions in the second and third quarters, and the largest number of interceptions in the fourth quarter (Fig 3). Over time, the proportion of different intercepted pest taxa has become increasingly balanced for all source continents. This reflects a movement of the Chinese port

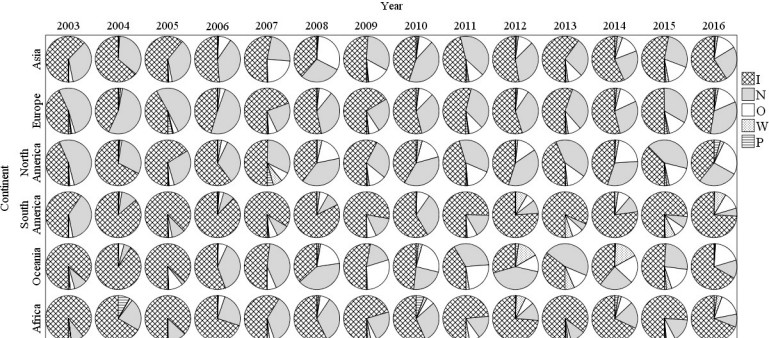

**Fig 2. Pie chart of wood packaging pest taxa intercepted from different continents from 2003 to 2016.** I = Insects, N = Nematodes, W = Weeds, P = Pathogens, O = Others (e.g. mites, spiders, mollusks).

quarantine away from an exclusive focus on insects and nematodes to a more comprehensive taxa inspection, thereby reducing the chance of unavoidable harm brought about by multiple pest taxa in wood packaging. This may reflect seasonal differences in commodity transport, insect activity, or quarantine sectors [34, 35].

Interceptions from each continent continued to increase throughout the 14-year period. Interception data points from Asia, Europe, and North America in the fourth quarter of 2014 were large outliers, primarily because of increased interceptions at the Jiangsu, Guangdong, and Shandong inspection stations during that period. The number of interceptions in the fourth quarter in 2014 was 13,893 records more than the 2009 to 2013 average. The interception of insects from the United States, Taiwan, Korea, Germany, Japan and other countries increased significantly, perhaps because of an increased volume of cargo in the fourth quarter and a high risk of epidemics in the cargo itself [36].

## Country of origin

The number of countries (regions) from which pests were intercepted on WPM at Chinese ports increased from 68 to 163 between 2003 and 2016, and the countries (regions) from which quarantine pests were intercepted increased from 26 to 107. The country (region) with the most frequent interceptions was Korea (12.98% of all records), followed by the United States (12.92%), Germany (11.84%), Taiwan (China) (10.64%), and Japan (6.64%). The most frequent insect interceptions were from Germany, Taiwan, and the United States, and the most frequent nematode interceptions were from Korea, the United States, and Germany. Other pests were most frequently intercepted from the United States, Korea, and Japan (Table 1).

When countries (regions) were sorted by the total number of intercepted pests (X2), interceptions were found to be concentrated in the first 45 countries (regions) (98.10% of the total records), and it is therefore reasonable to believe that these 45 countries (regions) are the main sources of WPM pest interceptions in China. Based on clustering analysis, these countries were divided into four categories (Fig 4). The first cluster groups Korea, the United States, Germany, Taiwan, and China with an extremely high values of X1 and X2. The second cluster includes seven countries (Vietnam, Philippines, Malaysia, Singapore, Thailand, Indonesia, and India) with a high values of X1, X2, and X4 and an extremely high ratio of intercepted quarantine pests (X3). The third cluster includes 19 countries such as Japan, Chile, UK, Italy, and Australia. Their values for the four statistical variables are moderate, and their data ranges are large. The last cluster contains fifteen countries including South Africa, Russia, Argentina,

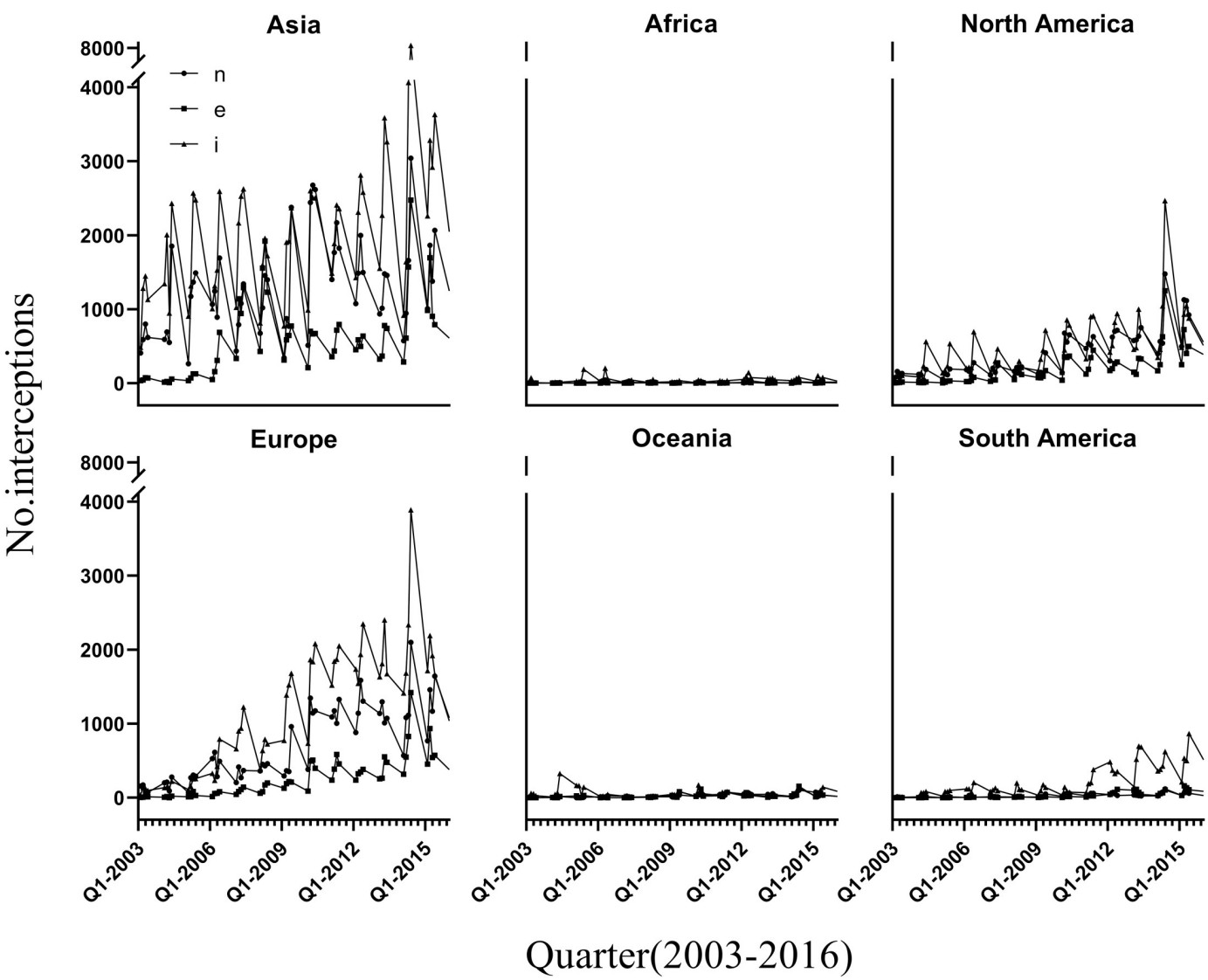

**Fig 3. Numbers of exotic pests intercepted from six continents during four quarters of each year from 2003 to 2016.** I = Insects, N = Nematodes, E = all other taxa.

Norway, and Denmark with low values of X1, X2, and X3, and their interceptions are primarily insects and nematodes. It should be noted that "low" numbers of pest species or intercepted pests are low only in comparison to other countries on the top 45 list. From an overall perspective, such low numbers of species or interceptions should not be underestimated [37].

Factor analysis of 251,877 interception data points produced a Kaiser–Meyer–Olkin (KMO) value of 0.726 (>0.5) and a highly significant result in Bartlett's sphericity test ($P$ <0.0001). Based on matrix eigenvalues and the cumulative variance contribution (S2 Table), the results were divided into two categories (S3 Table). The first category was dominated by nematodes, storage pests, and cerambycids (variance contribution rate of 69.71%) as the main pests in the imported WPM. The second one includes xylophagous insects. A correspondence analysis (Pearson's $\chi^2$ test; $\chi^2$ = 105,899, df = 170, $p$ <$2e^{-16}$) was performed on the countries with the top 35 composite scores (S4 Table). Xylophagous insects were clustered together, consistent with the results of the factor analysis, and Southeast Asian countries dominated. The

**Table 1. Top 20 countries of origin for pest taxa intercepted in China from 2003 to 2016.**

| Rank | Rank order by total interceptions (percent of total records) | Rank order by pathogen interceptions | Rank order by insect interceptions | Rank order by nematode interceptions | Rank order by interceptions of other taxa | Rank order by weed interceptions |
|---|---|---|---|---|---|---|
| 1 | Korea (12.98) | United States | Germany | Korea | United States | United States |
| 2 | United States (12.92) | Korea | Taiwan (China) | United States | Korea | Taiwan (China) |
| 3 | Germany (11.84) | Germany | United States | Germany | Japan | South Korea |
| 4 | Taiwan (China) (10.64) | Taiwan (China) | South Korea | Taiwan (China) | Germany | Chile |
| 5 | Japan (6.59) | Japan | Malaysia | Japan | Taiwan (China) | Germany |
| 6 | Hong Kong (China) (4.11) | Italy | Singapore | Hong Kong (China) | Hong Kong (China) | Australia |
| 7 | Italy (3.43) | France | Japan | Italy | Italy | Japan |
| 8 | Singapore (3.26) | United Kingdom | Hong Kong (China) | France | Singapore | Italy |
| 9 | Malaysia (3.21) | India | Thailand | Belgium | Thailand | Malaysia |
| 10 | Thailand (2.97) | Belgium | Indonesia | Singapore | Malaysia | Thailand |
| 11 | Indonesia (2.41) | Hong Kong (China) | Italy | Spain | Indonesia | Indonesia |
| 12 | France (2.37) | Turkey | India | Netherlands | France | India |
| 13 | India (2.02) | Singapore | Chile | United Kingdom | Belgium | Singapore |
| 14 | Belgium (1.7) | Netherlands | France | India | Australia | Belgium |
| 15 | Chile (1.68) | Indonesia | Brazil | Malaysia | Netherlands | Canada |
| 16 | Netherlands (1.47) | Thailand | Belgium | Thailand | United Kingdom | France |
| 17 | Brazil (1.47) | Australia | Netherlands | Sweden | India | Netherlands |
| 18 | United Kingdom (1.42) | Spain | United Kingdom | Canada | Chile | United Kingdom |
| 19 | Spain (1.10) | Vietnam | Vietnam | Brazil | Brazil | Saudi Arabia |
| 20 | Australia (0.98) | Malaysia | Philippines | Australia | Vietnam | Hong Kong (China) |

most interceptions of Scolytidae were associated with Singapore, whereas those of Platypodidae and Bostrichidae were associated with Thailand and Malaysia. Nematode interceptions were mainly from Korea, Japan, Australia, Mexico, and the United States. Storage pest interceptions were mainly from Chile, Brazil, Russia, Germany, and other European countries. The

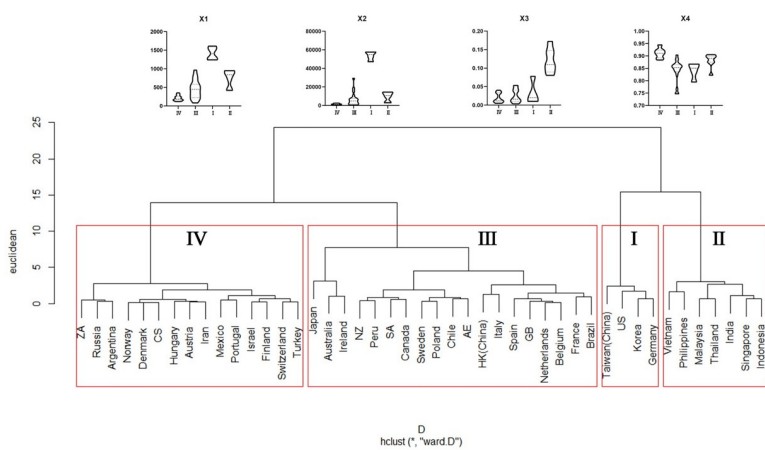

**Fig 4. Cluster dendrogram of the main countries (regions) from which pests are intercepted on WPM in China.** X1 = the total species of intercepted pests. X2 = the total number of intercepted pests. X3 = the interception rate of quarantine pests (quarantine pests intercepted/X2). X4 = the interception rate of insects and nematodes (insects and nematodes intercepted/X2). US = United States, ZA = South Africa, CS = Czech Republic, HK = Hong Kong, GB = United Kingdom, NZ = New Zealand, SA = Saudi Arabia, AE = United Emirates.

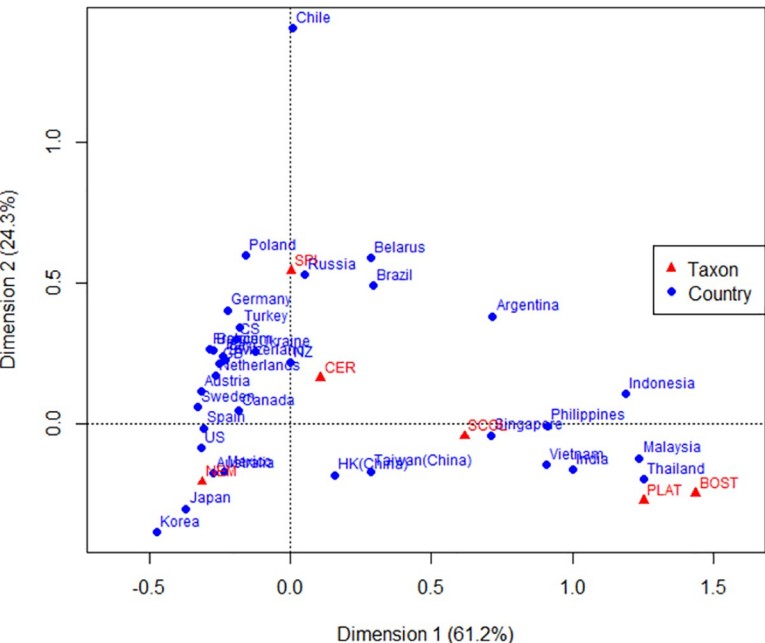

**Fig 5. Correspondence analysis of major insects and nematodes intercepted on WPM in China with their source countries (regions).**

Cerambycidae were closest to the origin of the coordinates, and no countries were nearby, and the number at Chinese ports was relatively low (Fig 5).

The distinct characteristics of WPM pest interceptions from different countries are closely related to the occurrence of these pests in their originating countries. This is partly related to their geographic locations, which are suitable for the growth and reproduction of specific pest species [38, 39]. In addition, WPM carry all kinds of pests, and their presence is related to WPM treatment and implementation standards in the originating countries (regions) [40, 41].

### Inspection stations

The WPM pest interception data from individual Chinese provinces, excluding Hong Kong, Macau, and Taiwan were statistically analyzed. Stations from 31 provinces uploaded records from 2003 to 2016, although Tibet's stations did not upload records. In terms of interception numbers, analysis of the top nine provinces showed that interceptions of insects and nematodes generally predominated (98.89% of all records), except for Hubei, where few nematode interceptions were recorded (Table 2).

Shanghai stations had the largest number of interceptions with a high number of insects (69.29% of Shanghai records) and other pests (24.3%) accounted for a large proportion, whereas nematodes and pathogens were very low with 5.46% and 0.1%, respectively. The proportion of nematodes (65.30% of Shangdong records) and other pests (14.68%) intercepted in Shandong was very high, whereas that of insects was low (17.42%). The proportion of insects and plants intercepted by the Guangdong Bureau was high with 81.86% and 2.52%, respectively and that of nematodes was low. In Tianjin and Liaoning, nematode interceptions predominated (75.87%) and (69.20%). Clearly, the proportion of different taxa intercepted varies among provinces.

Stations in China's coastal provinces are the most important area for WPM interceptions (98.7% of all records) (Fig 6). This probably reflects the development of economic trade in the

**Table 2. The top nine provincial stations for WPM pest interceptions from 2003 to 2016 in China.**

| Inspection station | Pathogens | Insects | Nematodes | Others | Weeds | Station Total | % of Total |
|---|---|---|---|---|---|---|---|
| Shanghai | 19 | 101,730 | 8021 | 35,722 | 1331 | 146,823 | 31.61 |
| Jiangsu | 901 | 55,542 | 57,996 | 4346 | 3018 | 121,803 | 26.22 |
| Guangdong | 764 | 60,921 | 5209 | 5648 | 1876 | 74,418 | 16.02 |
| Shandong | 1119 | 12,483 | 46,787 | 10,521 | 739 | 71,649 | 15.42 |
| Zhejiang | 520 | 6280 | 11,520 | 529 | 115 | 18,964 | 4.08 |
| Tianjin | 3 | 3225 | 10,821 | 191 | 23 | 14,263 | 3.07 |
| Liaoning | 18 | 919 | 4017 | 753 | 98 | 5805 | 1.25 |
| Fujian | 164 | 3219 | 525 | 127 | 21 | 4056 | 0.87 |
| Hubei | 653 | 841 | 7 | 24 | 35 | 1560 | 0.34 |
| Total | 4161 | 245,160 | 144,903 | 57,861 | 7256 | 459,341 | 98.89 |

coastal provinces [42], where transport carriers of intercepted WPM cargo are mainly freighters (55.51%) and containers (26.27%). Data from the six provinces with the largest total numbers of interceptions clearly demonstrate that interceptions in the southern and northern

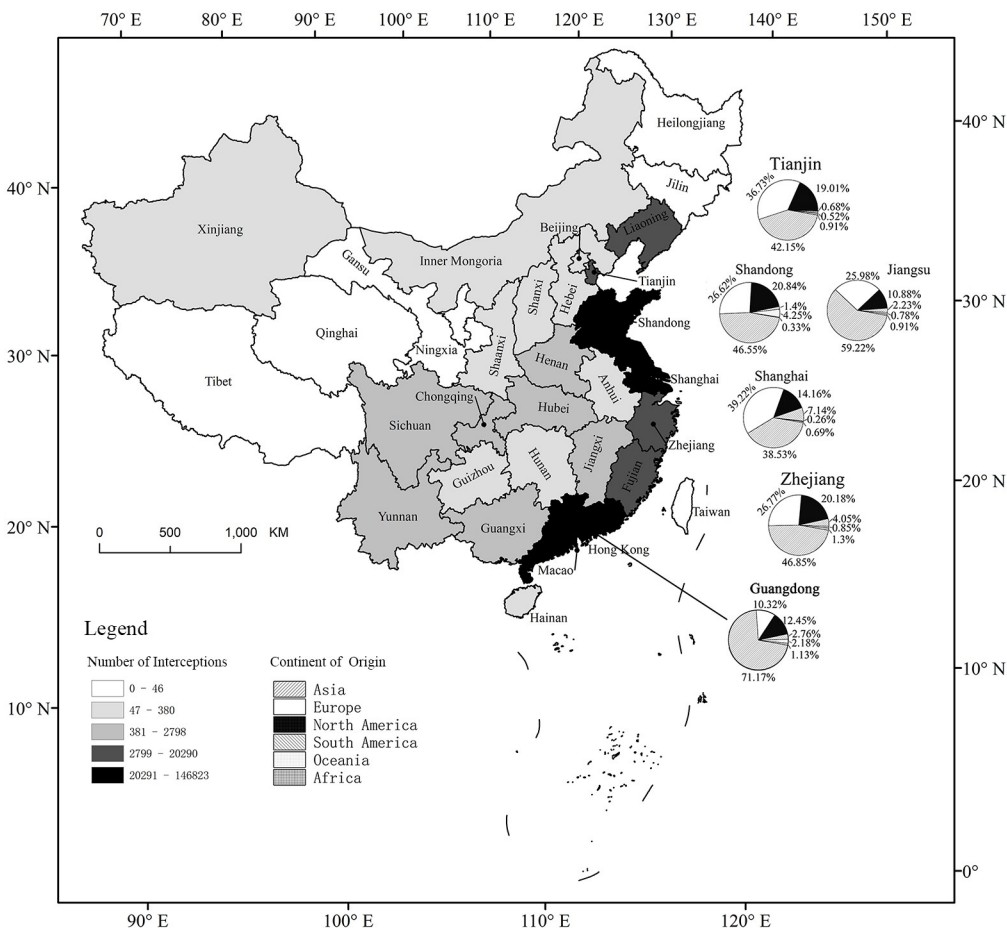

**Fig 6. Number of exotic pests intercepted at inspection stations in each province and the percentage of exotic pests from major continents in the six provinces with the highest number of interceptions.** CER = Cerambycidae, SCOL = Scolytidae, PLAT = Platypodidae, BOST = Bostrichidae, NEM = Nematode, SPI = Storage pests, US = United States, ZA = South Africa, CS = Czech Republic, HK = Hong Kong, GB = United Kingdom, NZ = New Zealand, SA = Saudi Arabia.

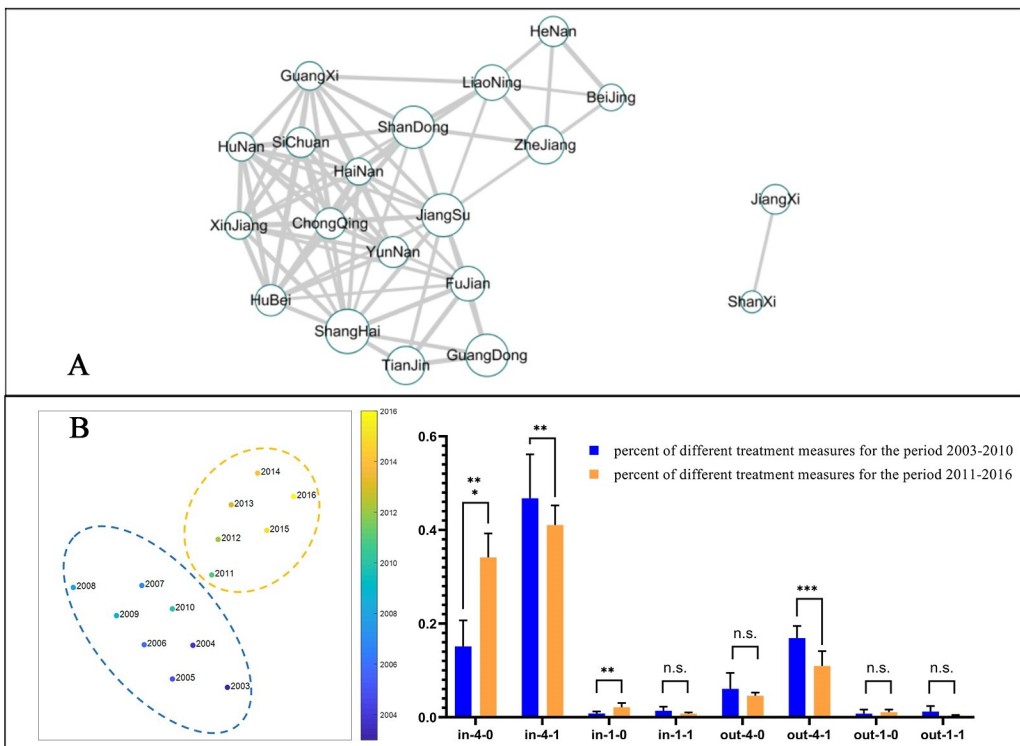

**Fig 7. A network diagram of "survival status–quarantine status–treatment measures" and its statistical analysis by year.** (A) A "survival status–quarantine status–treatment measures" network diagram for provincial ports. The size of the circle represents the number of interceptions, the thickness of the line represents the size of the correlation, and lines are only produced for correlations >0.8. (B) A "survival status–quarantine status–treatment measures" clustering analysis by year. Quarantine status is denoted by 1 for quarantine pests and 4 for non-quarantine pests. Treatment measures are denoted by "out" for return, destruction, and sealing and "in" for inspection and quarantine supervision, fumigation, and pest control treatment. Survival status is denoted by 1 for living and 0 for dead. Together these three categories give rise to eight combinations. *** P <0.01, ** P <0.05, n.s. P >0.05, Student's *t*-test.

stations are dominated by pests from North America, Asia, and Europe. South America's pests are mainly intercepted in the southern stations, and Oceania's pests are mainly intercepted in the northern stations.

## Treatment of cargo intercepted by stations

The treatment of cargo intercepted by stations of the top 20 Chinese provinces based on number of interceptions was 99.9% of all records (Fig 7). Jiangxi's station has the most stringent treatment measures, and a relatively high proportion of cargo in which non-quarantine pests are found is destroyed or returned. Followed by Beijing, Henan, Zhejiang, and other provinces in which cargo with living non-quarantine pests is frequently destroyed and treatment measures are relatively strict. The most lenient treatment measures are found in Xinjiang's stations, where all intercepted cargoes are subjected to pest disposal treatment (Fig 7A).

The combination of "survival status–quarantine classification–treatment measures" was clustered by year into two categories, 2003–2010 (category I) and 2011–2016 (category II). Compared with category I, the number of pest disposal treatment measures for cargo with dead non-quarantine pests was significantly higher in category II. The number of destruction treatment measures for cargo with live non-quarantine pests was significantly lower, and the number of pest disposal treatments for cargo with dead quarantine pests was higher (Fig 7B).

All these results reflect the increasingly standardized and scientific measures taken by Chinese customs inspection stations to deal with intercepted pests on WPM cargo [43].

## Conclusions

The WPM interception pest record is a part of the CPIN. It is still a valuable historical record of a range of pests entering China and their arrival pathways. Insects and nematodes are easily carried by WPM, thereby promoting their invasion and spread. Goods from countries with the highest total number of interceptions require great attention at Chinese customs ports. It is also recommended that more targeted WPM inspection measures be taken for relevant countries in the future by increasing the sampling volume and strengthening the follow-up supervision for countries where the interception rate of quarantine pests is high.

In conclusion, Pest Risk Analysis should be conducted to clarify the hazards and invasion risks of relevant pests in advance mainly for countries/regions with a high category and number of intercepted pests.

The quarantine pests signaled in this work should receive special attention to improve the relevance and validity of the inspections. It is also suggested that provincial stations develop more detailed treatment measures to ensure economic development and effectively intercept exotic pests.

## Supporting information

**S1 Table. Definitions of terms.**
(DOCX)

**S2 Table. Matrix Eigenvalue and cumulative variance contribution rate of factor analysis.**
(DOCX)

**S3 Table. Rotated factor loading matrix.**
(DOCX)

**S4 Table. The factor scores and rankings of original countries (regions) of intercepted entry WPM pests.**
(DOCX)

**S1 Data.**
(XLSX)

## Acknowledgments

We thank the Chinese Academy of Inspection and Quarantine (CAIQ) for supporting us with some data and materials.

## Author Contributions

**Conceptualization:** Juan Shi.

**Data curation:** Ke Chen.

**Writing – original draft:** Jiaqiang Zhao, Ke Hu.

**Writing – review & editing:** Juan Shi.

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
