## [Decision Letter · Decision Letter 0]

2 Jun 2021

PONE-D-21-14207

Quarantine Supervision of Wood Packaging Materials (WPM) at Chinese Ports of Entry: 2004–2016

PLOS ONE

Dear Dr. Zhao,

Thank you for submitting your manuscript to PLOS ONE. After careful consideration, we feel that it has merit but does not fully meet PLOS ONE’s publication criteria as it currently stands. Therefore, we invite you to submit a revised version of the manuscript that addresses the points raised during the review process.

We look forward to receiving your revised manuscript.

Kind regards,

Ramzi Mansour

Academic Editor

PLOS ONE

Journal Requirements:

4. We note that Figure 6 in your submission contain map images which may be copyrighted. All PLOS content is published under the Creative Commons Attribution License (CC BY 4.0), which means that the manuscript, images, and Supporting Information files will be freely available online, and any third party is permitted to access, download, copy, distribute, and use these materials in any way, even commercially, with proper attribution. For these reasons, we cannot publish previously copyrighted maps or satellite images created using proprietary data, such as Google software (Google Maps, Street View, and Earth). For more information, see our copyright guidelines: http://journals.plos.org/plosone/s/licenses-and-copyright.

4.1.    You may seek permission from the original copyright holder of Figure 6 to publish the content specifically under the CC BY 4.0 license. 

4.2.    If you are unable to obtain permission from the original copyright holder to publish these figures under the CC BY 4.0 license or if the copyright holder’s requirements are incompatible with the CC BY 4.0 license, please either i) remove the figure or ii) supply a replacement figure that complies with the CC BY 4.0 license. Please check copyright information on all replacement figures and update the figure caption with source information. If applicable, please specify in the figure caption text when a figure is similar but not identical to the original image and is therefore for illustrative purposes only.

Reviewers' comments:

Reviewer's Responses to Questions

**Comments to the Author**

1. Is the manuscript technically sound, and do the data support the conclusions?

Reviewer #1: Yes

Reviewer #2: Yes

Reviewer #3: Yes

2. Has the statistical analysis been performed appropriately and rigorously? 

Reviewer #1: Yes

Reviewer #2: Yes

Reviewer #3: Yes

3. Have the authors made all data underlying the findings in their manuscript fully available?

Reviewer #1: Yes

Reviewer #2: Yes

Reviewer #3: Yes

4. Is the manuscript presented in an intelligible fashion and written in standard English?

Reviewer #1: Yes

Reviewer #2: Yes

Reviewer #3: No

5. Review Comments to the Author

Reviewer #1: I read the manuscript by Zhao et al. with interest. The paper aims to analyse interception data for Wood Packaging Materials (WPM) at Chinese Ports of Entry from 2003 to 2016, they discuss the issues reflected in the database, with a view to providing a reference for future work by customs officers and researchers.

In general the topic is relevant and the tests carried out can have important implications in the background dataset for managing future WPM quarantine of incoming goods at Chinese ports and will help to prevent foreign pests from entering China on WPM through international trade, however there are some improvements to be made:

Comment 1: If you have the data, looking at different species of pests (insects, mites, etc,….) might be a beneficial addition to the paper.

Comment 2: I suggest to the authors to report the geographic coordinates of the study areas.

Comment 3: figure 7: Explain which type of pesticides were used, how the number of treatments required and the choice of pesticides are established. I suggest authors to motivate the choice of pesticides used in the study. On the basis of what they were chosen? Are they the ones most used by companies? Are there any relationships between them? Please improve this aspect.

Comment 4: How are the pests ( insects, mites, nematodes, etc,…) sampled? Identification is done by which method? Please provide this information.

Reviewer #2: The paper is well written and give useful information on quarantine supervision system. There is an accurate analysis of the results but probably lacks of considerations on the quarantine pests and their countries of origin. Some figures need corrections in order to make them more readable.

All specific comments and suggestions are stated in the attached file.

Reviewer #3: Authors give information about Wood Packaging Material (WPM) in China from entry pathways. They conclud that insects and nematods are the most pests in the WPM.

Correction and comments are inserted in the text.

6. PLOS authors have the option to publish the peer review history of their article (what does this mean?). If published, this will include your full peer review and any attached files.

Reviewer #1: No

Reviewer #2: No

Reviewer #3: **Yes: **Ben Jamaa Mohamed Lahbib

---

## [Author Response · Author response to Decision Letter 0]

13 Jul 2021

On behalf of my co-authors, we thank you very much for giving us an opportunity to revise our manuscript, we appreciate editor and reviewers very much for their positive and constructive comments and suggestions on our manuscript ，I have uploaded the specific feedback in the attachment "Respnse to Reviewers", please download it to view.We tried our best to improve the manuscript and made some changes in the manuscript. Thank you and I wish you all good health and success in your work.

---

## [Editor Report · Decision Letter 1]

23 Jul 2021

Quarantine Supervision of Wood Packaging Materials (WPM) at Chinese Ports of Entry from 2003–2016

PONE-D-21-14207R1

Dear Dr. Zhao,

We’re pleased to inform you that your manuscript has been judged scientifically suitable for publication (BUT, please see and apply ADDITIONAL EDITOR COMMENTS below) and will be formally accepted for publication once it meets all outstanding technical requirements.

Kind regards,

Ramzi Mansour

Academic Editor

PLOS ONE

Additional Editor Comments:

The following revisions should be made by the authors on the PROOFS of their accepted article:

L3 (title):  replace  "from 2003-2016"   with   "from 2003 to 2016"

L15-16 (Abstract):  replace  "on the species and origins of these pests will help"    with   "on all concerned pest species and their origin will help"

L58:   delete  "of the wood"  avoiding its repetition

L72-75:   "et al."  should be italicized and the reference number should be placed just after "et al." as follows:   change all the sentence to  "and Mccullough et al. [21] analyzed interception data for nonindigenous plant pests for 17 years and found that within specific commodity pathways, richness of the pest taxa generally increased linearly with the number of interceptions."

L78-80:   change the sentence to  "Xia et al. [23] analyzed the annual trends, population types, and interception frequencies of quarantine pests intercepted on imported wood packaging in Shandong Province, China".

L87:  replace  "the types"   with  "the living organism groups"

L89:  replace  "the main of this work is to use WPM interception"   with  "the main purpose of the present study was to use WPM interception"

L106:  replace "Latin name"  with  "scientific name"

L156:  replace "mainly related to"  with  "mainly due to"

L167:  add  " respectively, "   before  "followed by"

L175:  delete "Abbreviations are:"

L179:  delete the comma after  "origin"

L179-180:  delete "which may be caused abandoned items or missing labels"

L215:  delete "Abbreviations are:"

L218:  replace  "Fig 3. The numbers of"   with  "Fig 3. Numbers of"

L220:   delete  "Abbreviations are:"

L226:  delete "(region)"

L259-260:  delete "Abbreviations are:"

L338:  delete "Abbreviations are:"

L379-380:  replace "*p*"   with   "*P*"

L383:  change to   " of the CPIN. It is still a "
---

## [Editor Report · Acceptance letter]

27 Jul 2021

PONE-D-21-14207R1 

Quarantine Supervision of Wood Packaging Materials (WPM) at Chinese Ports of Entry from 2003 to 2016. 

Dear Dr. Zhao:

I'm pleased to inform you that your manuscript has been deemed suitable for publication in PLOS ONE. Congratulations! Your manuscript is now with our production department. 

Kind regards, 

on behalf of

Dr. Ramzi Mansour 

Academic Editor

PLOS ONE